# Zinc Adequacy Is Essential for the Maintenance of Optimal Oral Health

**DOI:** 10.3390/nu12040949

**Published:** 2020-03-30

**Authors:** Anne Marie Uwitonze, Nkemcho Ojeh, Julienne Murererehe, Azeddine Atfi, Mohammed S. Razzaque

**Affiliations:** 1Department of Preventive & Community Dentistry, University of Rwanda College of Medicine & Health Sciences, School of Dentistry, KK 737 St, Kigali, Rwanda; 2Faculty of Medical Sciences, University of the West Indies, Cave Hill Campus, Bridgetown BB11000, Barbados; 3Department of Pathology, Virginia Commonwealth University, Richmond, VA 23284, USA; 4College of Advancing & Professional Studies (CAPS), University of Massachusetts Boston (UMB), Boston, MA 02125, USA; 5Department of Pathology, Lake Erie College of Osteopathic Medicine, Erie, PA 16509, USA

**Keywords:** zinc, oral health, oral tumor, periodontal diseases, oral mucositis

## Abstract

Zinc, a metal found in the Earth’s crust, is indispensable for human health. In the human body, around 60% of zinc is present in muscles, 30% in bones, and the remaining 10% in skin, hair, pancreas, kidneys and plasma. An adequate zinc balance is essential for the maintenance of skeletal growth, development and function. It is also necessary for basic cellular functions including enzyme activation, cell signaling and energy metabolism. Inadequate zinc status is associated with a wide variety of systemic disorders including cardiovascular impairment, musculoskeletal dysfunctions and oromaxillary diseases. In this article, we briefly discuss the role of zinc deficiency in the genesis of various oromaxillary diseases, and explain why adequate zinc homeostasis is vital for the maintenance of oral and general health.

## 1. Introduction

The optimal level of zinc is important for the growth and development of human health [1]. In the human body, zinc is found in muscles (60%), bones (30%) and skin (5%) [2]. It has an array of functions including being involved in the activation of various enzymes and proteins [3], and zinc is contributing to the absorption of vitamin A, E, and folate [1]. Low levels of zinc can be associated with an increased chance of developing infections and degenerative pathologies [1]. Zinc also plays an important role in the psychosocial functioning of human behavior [1]. 

In the oral cavity, zinc is found in saliva, dental plaque and in the hydroxyapatite of the dental enamel [4]. It contributes to healthy teeth formation [4], and is used in mouth rinses and toothpaste due to its important role in the prevention of plaque and dental calculus formation [2]. Zinc also contributes to the reduction of halitosis in the mouth [2]. It has been implicated in the composition of dental biomaterials and orthodontic materials, due to its properties for enhancing immunity, as well as its effects on cell division and skeletal development [5]. Clinical trials have demonstrated that zinc ions decrease the rate of enamel demineralization [2]. The concentration of zinc in enamel surface ranges between 430 to 2100 parts per million (ppm) and it is deposited mostly before tooth eruption [6]. Zinc is important for maintaining periodontal health because of its local and immunological effect on oral soft tissues. As with other micronutrients that fall into the category of minerals that are needed in quantities of <100 mg/day [1], the recommended daily allowance for zinc ranges between 2 to 13 mg/day depending on the stage of life and sex of the individual, with the upper limit for zinc being set at 40 mg/day. 

## 2. Zinc Homeostasis

When zinc is ingested orally, 25–66% is absorbed through the small intestine especially from the jejunum and ileum. There is no specific zinc store in the body as it is present in all body tissues and fluids, including blood cells, pancreas, retina, prostate, kidneys, lungs, skin, liver, brain, the gastrointestinal tract, choroid of the eye, and the heart [7,8], with muscles and bones having the highest zinc content. Men typically have slightly higher body zinc content (2.5 gm) compared to women (1.5 gm).

Serum, where zinc is bound to proteins such as albumins (57%), α2 globulin (40%), transferrin and amino acids (3%), is responsible for the distribution of zinc throughout the body, even though it represents only less than 2% of the total body content. The intracellular content of zinc, present in the cytosol, organelles such as nuclei and vesicles among others, represents more than 95% of the total body content [9]. There is a delicate zinc homeostatic control at the cellular level that avoids excessive zinc accumulation [7]. It is mediated by 14 protein members of the zinc importers (Zip) family which import zinc into the cytosol and 10 protein members of the zinc exporters (ZnT) family which transport zinc out of the cytosol; they also regulate the distribution of zinc in the intracellular organelles such as Golgi apparatus, the endoplasmic reticulum and the mitochondria [10] (Figure 1).

Another way of maintaining intracellular zinc homeostasis is through metallothioneins (MT) which are zinc-binding proteins that can bind up to seven zinc ions each and act to buffer cellular zinc [10]. Additionally, many mammalian cells contain zincosomes, which are membrane-bound vesicles that sequester a large amount of zinc and release it when needed [7]. Zinc is eliminated from the body through the kidneys where the amount depends on the level of starvation and muscle catabolism, through the skin depending on the level of exercise and the ambient temperature, and through the intestines depending on zinc intake. Zinc is 16% more concentrated in blood serum than plasma due to dilution factor, hemolysis as well as platelet disintegration [4].

## 3. Sources and Recommendation of Zinc

The content of zinc in foods depends on its content in the soil. Table 1 lists some of the foods containing zinc. Meat and poultry-based products and seafood, including lamb, liver, beef, rabbit, chicken, oyster and lobster are the best sources of zinc. They should be eaten together with vegetables for better absorption of zinc [9]. Moreover, whole grains such as black rice, black sesame, bread, and noodles are also good sources of zinc. Vegetables such as soy foods, mushroom, edible fungi, celery, legumes such as beans, peas, and lentils, nuts, and seeds such as sunflower seeds and almonds, among other food sources, all contribute to zinc consumption [4]. Although tubers, cereals, and legumes, which are consumed in African countries, contain a good amount of zinc, the concomitant presence of phytates, lignin, and fibers, hinders its bioavailability. The presence of casein and calcium in cow milk and phytate in soya milk may reduce the absorption of zinc from the diet. However, this does not apply to breastmilk where zinc is well-absorbed [9]. This indicates that the habit of some individuals of drinking milk while or immediately after eating is not helpful to the body and should be consumed 30–60 min after a meal. Some portion of zinc is lost during the process of refining cereals [4]. Largely, fruits and vegetables are typically poorer sources of zinc [4] although dates, pomegranates, berries, and avocadoes have a good amount of zinc. 

Recommended Daily Intake (RDI) and Recommended Daily Allowance (RDA) numbers are estimated amounts that are needed to prevent people from manifesting signs and symptoms of deficiency. The zinc RDI at different levels of life is listed in Table 2. RDA of zinc varies from 2 mg in infants to 13 mg in adults. Physiological requirements for zinc are different for adult males and females. It is very important to differentiate between zinc intake and absorption because fibers and phytates can inhibit zinc absorption, thereby leading to reduced availability, even though dietary intake is adequate [12].

## 4. Zinc Deficiency

Studies have shown that zinc deficiency is very common. It can be caused by inadequate dietary intake, poor absorption and increased loss [13]. Zinc deficiency is more severe in the developing world, where 2.5 billion children are deficient in zinc and it is a greater health risk than zinc intoxication [7]. According to Zofkova et al. (2017), one-quarter of the world’s population is deficient in zinc [14]. Worldwide, 800,000 children die annually from zinc deficiency [12] and especially from diarrhea-related diseases and respiratory infections, both of which increase body losses or body requirements of zinc [9]. Zinc deficiency can be primary or secondary. The three most prominent among them are: (**1**) diets which are poor in zinc; (**2**) a rare autosomal recessive condition called acrodermatitis enteropathica due to mutations in the gene (SLC39A4) which codes the zinc transporter protein, ZIP4; (**3**) exclusively breast-fed babies as a result of mutations in the SLC30A2 (zinc T-2), a gene which is responsible for transferring zinc from serum to breast milk [4]. 

Various disorders such as sickle cell disease, diabetes, cirrhosis of the liver, burns and other chronic disorders; certain foods including low mineral purified foods and those containing additives with chelating activities; poor sanitation, socioeconomic status, and other factors can lead to secondary or conditioned zinc deficiency. Zinc deficiency can be associated with some malignancies, multiple sclerosis, sepsis, senility, Alzheimer’s disease, Parkinson’s disease onset and progression [9,15,16]. Medications such as ethambutol, anticonvulsants, penicillamine, iron supplements, laxatives, and antacids are also known to contribute to zinc deficiency [4,7,13]. 

Environmental factors such as steel production, burning of coal or waste can deplete the soil of zinc making it loose in the atmosphere. Some groups of people are at a higher risk of zinc deficiency including developing fetuses, pregnant women, infants, children, adolescents, elderly individuals, and women on a weight-reducing diet. These individuals all require high amounts of zinc. Studies have revealed that alcohol addiction is also a contributing factor to zinc deficiency as it decreases zinc absorption and increases zinc urinary secretion [9]. Furthermore, tannin contained in coffee can also interfere with zinc absorption [12]. 

## 5. Role of Zinc in Oral Health and Disease

In the oral cavity, zinc is naturally present in various sites such as saliva, dental plaque and dental hard tissues. Supplementation with zinc is effective against different oral diseases, for example, gingivitis, periodontitis, halitosis, and others, while zinc deficiency has been associated with poor oral and periodontal health [17,18,19,20]. In a rat study that evaluated the effects of zinc deficiency on the oral tissues of rats, animals were fed a zinc-deficient or zinc-containing diet. Clinical findings indicated a lower gingival index in zinc-containing rats (*p* = 0.001) and a higher mean pocket depth in zinc-deficient rats. Zinc-deficient rats also displayed aphthous on the floor of the mouth compared to zinc-containing rats. The authors concluded that oral and periodontal health was better in zinc-containing rats than in zinc-deficient rats [20]. In another study wherein patients used mouthwash containing both cetylpyridinium chloride (CPC) and 0.28% zinc lactate, CPC alone or no CPC for up to six weeks, a greater reduction in plaque and gingival indices was seen with the mouthwash containing CPC and zinc compared to those with CPC alone or no CPC [18]. Zinc-based salts such as zinc oxide, zinc chloride, zinc citrate, zinc lactate, and zinc sulfate are used in various oral health care products such as toothpaste and mouth rinses to control plaque formation and inhibit calculus deposition. Zinc salts inhibit volatile sulfur compound production thus helping to reduce halitosis [17]. Its elevated concentration can be maintained in plaque and saliva for a long period, prolonging its beneficial effects [17]. Zinc deficiency has been associated with decreased taste and lingual trigeminal nerve sensitivities as well as reduced salivary flow [21,22].

Randomized controlled trials performed on rats by Seyedmajidi and colleagues (2014) and Ozler (2014) found that zinc deficiency may be harmful to the oral mucosa. Clinical findings showed that aphthous ulcer was higher in zinc-deficient rats especially on the floor of the mouth [20,23]. A similar study performed in 2007 by Orbak and colleagues found that alveolar mucosa was more affected by aphthous ulcer than other oral surfaces in zinc-deficient rats [24]. A case report on two siblings with acrodermatitis enteropathica showed that three days after birth, there were erythematous, erosive and crusted lesions on their mouths and tongues [25]. Mehdipour and colleagues in their study found that zinc mouthwash was effective in the wound healing process and treatment of recurrent aphthous stomatitis [26]. In a recent retrospective study in which patients with recurrent aphthous stomatitis received replacement treatment for the deficiency, the authors concluded that zinc deficiency should be considered and investigated in patients suffering from this condition [27]. In a double-blinded placebo-controlled study, in which patients were given 150 mg zinc sulfate, 50 mg dapsone or 250 mg glucose as a placebo, twice daily, the zinc sulfate treatment was found to have a much more rapid and sustainable action on aphthous ulcer than dapsone [28]. Besides these studies on the aphthous ulcer, zinc deficiency was found to be associated with the pathogenesis of other common oral mucosa diseases such as oral lichen planus, burning mouth syndrome, atrophic glossitis, and xerostomia. The researchers concluded that zinc supplementation may be a good treatment for oral mucosa diseases [29]. It is, however, important to mention that despite the association between zinc deficiency and aphthous ulcer in human studies, whether such association with zinc inadequacy is causative or a consequence, is an area that needs further experimental validation. 

## 6. Effects of Zinc on Dental Caries and Periodontal Tissues

A systematic review done by Fatima and coworkers highlighted that zinc is more concentrated in the superficial layer of the enamel, making it harder than the subsurface and more resistant to dental caries [17]. Studies based on synthetic apatite showed that zinc competes with calcium to attach on apatite, thereby making the hydroxyapatite more resistant to acid dissolution. The incorporation of zinc reduced the solubility of the carbonated and the noncarbonated apatite and its solubility to acid [17]. Toothpaste containing both fluoride and zinc has been proven to be more effective in preventing dentine demineralization and promoting remineralization, when compared to those containing fluoride alone. An increased concentration of zinc was also found to inhibit bacterial growth by targeting glycolytic enzymes especially in streptococcus mutants, inhibit acid production in dental plaque and inhibit the formation of plaque, all of which contribute to dental caries occurrence [17]. In 2009, Uckardes and colleagues performed a double-blinded randomized controlled trial on low socioeconomic status, healthy primary school children. One group was given 15 mg/day zinc supplement and the control group was given a placebo, for five days a week for 10 weeks. The results of the study revealed that the plaque index improved statistically only in the zinc group. The researchers concluded that zinc supplementation can contribute to dental caries prevention [30]. 

Zinc supplementation is effective in preventing gingival disease; it can fight against Fusobacterium nucleatum and Prevotella intermedia, which cause gingivitis. Zinc also inhibits proteases produced by Porphyromonas gingivalis [17]. Zinc oxide nanoparticles were found to disrupt the cell membrane and oxidative stress in Campylobacter species as well as in other gram-positive and gram-negative bacteria found in gingivitis [17]. A six-month randomized clinical study comparing two dentifrices, one containing fluoride and zinc and the other containing fluoride alone, revealed that the zinc-based dentifrice provided a meaningful clinical reduction in gingivitis and dental plaque [19]. A study performed by Hong and colleagues demonstrated that A20, an amino-acid protein with zinc finger C-terminal domains, may be a potential element in preventing inflammatory bone loss diseases such as periodontitis due to its anti-inflammatory and anti-osteoclastogenic effects [31]. Conversely, researchers have highlighted an association between zinc deficiency and gingivitis [17]. In animal studies, plaque index and gingival index were found to be low in rats fed with a zinc component than in controls and the mean pocket depth was higher in zinc-deficient rats [20,24]. 

## 7. Effects of Zinc Deficiency on Oral Malignancy

A study that investigated the role of zinc, along with other trace elements including copper and iron in the premalignant condition, oral submucous fibrosis (OSMF), was performed on fifty patients with OSMF and fifty healthy controls [32]. The level of serum zinc was significantly lower in patients (73.48 ± 24.21) than in controls (119.48 ± 52.78). Lower serum iron levels and higher copper levels in case subjects compared to normal controls were also found. The authors indicated that: (**1**) the low zinc levels observed may be attributed to the malignant cells needing more zinc, thus depleting the serum zinc levels; (**2**) the areca nut, which has a high copper content (302 nmol/g) and is the primary etiological factor for the development of OSMF, was the main cause for the increase in serum copper levels seen in the Indian patients recruited in the study who had a positive history of chewing the areca nut and; (**3**) the high copper levels may have contributed to the low zinc levels because of negative interaction between the two trace elements [32]. The authors concluded that based on their findings as well as findings from other studies, the level of zinc (and other trace elements) can be a potential diagnosis and prognosis marker of oral submucous fibrosis; however, further studies are needed for validation [32]. In support of these findings, meta-analysis and a systematic review of all studies relating to zinc and other trace elements including, copper and iron in patients with and without OSMF were conducted, and it was found that zinc level tested in serum, plasma or saliva was lower in patients with OSMF and correlated well with disease progression. These studies also reported an increase in copper levels and a decrease in iron levels with OSMF patients [33,34,35]. Reduced zinc levels may be caused by increased copper levels, contributed to in part by areca nut chewing, which in turn overstimulates collagen to be produced by oral mucosal fibroblasts by upregulating lysyl oxidase enzyme, which is important in the crosslinking of collagen and elastin [33]. This leads to the formation of fibrotic bands in the buccal mucosa of OSMF patients. The overproduction of collagen may result in the reduced plasma iron levels seen in OSMF patients [35]. The authors concluded that based on results from the studies, alongside clinical features, zinc may be an additional biomarker for development, diagnosis and prognosis of oral submucous fibrosis and other premalignant lesions, but that further studies were required [33,34,35].

Zinc exhibits proapoptotic, anti-inflammatory and antioxidant activities and is a component of many DNA repair proteins. Zinc induces activation of superoxide dismutase (SOD), an antioxidant enzyme, which inhibits reactive oxygen species production, hence protecting DNA from damage. Zinc also plays an important role in the production of metallothionein protein which protects against oxidative damage to cells [35]. The SOD enzyme was found to be reduced in OSMF patients and the increased copper levels, seen in these patients, increase free-radical production leading to oxidative stress [35]. Therefore, zinc deficiency renders cells more susceptible to oxidative stress and affects DNA repair. Zinc deficiency also causes upregulation of cyclooxygenase-2 (COX-2) which inhibits apoptosis while enhancing cells to proliferate, thus contributing to the malignant transformation of OSMF to oral cancer [35]. Hence, an adequate zinc level is important to prevent the transformation of the lesion into malignancy due to its involvement in DNA repair and the regulation of apoptosis and defense responses to oxidative stress. Researchers have also reported that proper nutrition could reduce the number of deaths from cancer. They highlighted that zinc may be of particular importance in host defense against the initiation and progression of cancer [34,36].

In a randomized cross-sectional study in which serum levels of trace elements including zinc, copper, iron, magnesium and calcium were investigated in patients with oral squamous cell carcinoma, OSMF, or normal patients to determine disease progression and association with areca nut and betel quid chewing habits, the levels of zinc along with iron and magnesium were found to be significantly lower in patients with oral squamous cell carcinoma and OSMF [37] whereas copper level was increased. The authors concluded that assessing the serum level of zinc, as well as other trace elements, may serve as a useful biomarker of oral squamous cell carcinoma and progression from premalignant to malignant oral lesions [37]. Primarily, it is therefore important to check zinc levels in patients with suspicious lesions for early detection and treatment initiation for oral squamous cell carcinoma which could lead to better prognosis [38]. In an animal study, the pathogenesis of induced tumor growth in submandibular salivary glands of albino rats was found to be modified and retarded when zinc concentration in the drinking water reached 250 ppm. As zinc concentration increased in the drinking water, the carcinomatous epithelium decreased and the inflammatory response was more marked [39].

None of the aforementioned studies on OSMF or oral squamous cell carcinoma patients reported on dietary intake data. Therefore, the effect of diet on zinc deficiency in these patients is not known. It cannot be ruled out, however, that the restricted mouth opening in OSMF patients can lead to reduced food intake and hence zinc or other trace element deficiency. In addition, few studies have reported the effect of supplementation of trace elements to treat or correct zinc deficiency in OSMF patients and none have measured zinc level or other trace elements after the intervention. Hence, further investigation in this area is needed [35].

## 8. Effects of Zinc on Oral Mucositis

Chemotherapy and radiotherapy are used to treat cancer. However, their cytotoxic effects can cause a severe condition called oral mucositis. When they are used concurrently, the incidence of oral mucositis can be as high as 90%. Oral mucositis is an acute inflammation of the oral mucosa following exposure to either chemotherapy and/or radiotherapy. It harms the quality of life as patients have difficulty eating, drinking, and swallowing due to mouth pain [40]. Several studies have been done to verify the possible effects of zinc sulfate in the prevention of oral mucositis or alleviation of symptoms. However, results are still controversial. 

In 2018, a meta-analysis of five randomized controlled trials did not find any beneficial effects of oral zinc sulfate on chemotherapy-induced oral mucositis. They found that oral zinc sulfate may not produce any clinical effects either in preventing or reducing the incidence, alleviating pain intensity, or reducing the severity of chemotherapy-induced oral mucositis [41]. In a randomized trial including leukemia patients undergoing chemotherapy to investigate whether zinc prevented chemotherapy-induced oral mucositis, supplementation with zinc sulfate for a month was not found to be associated with decreased pain intensity [42]. A systematic review and meta-analysis of four randomized control trials that investigated the potential effects of zinc sulfate in the prevention of radiation-induced oral mucositis among patients suffering from head and neck cancers undergoing radiotherapy, revealed that zinc sulfate may not have the benefit of prophylaxis against radiation-induced oral mucositis in these patients [43]. Similar findings were obtained in other studies focused on the potential effects of zinc sulfate in the prevention of radiation-induced oral mucositis among patients suffering from head and neck cancers undergoing radiotherapy [42,43]. In a double-blind, randomized, placebo-controlled study in which patients were given 440 mg zinc sulfate daily or placebo for 3 weeks, Mansouri and colleagues did not find any significant association between zinc sulfate and prevention, reduction in intensity and duration of oral mucositis among patients undergoing high-dose chemotherapy [44]. In contrast, results obtained from another randomized clinical trial, placebo-controlled, triple blinded study on leukemia patients undergoing chemotherapy revealed that the incidence of chemotherapy-induced oral mucositis was 2.1 times higher in the control group compared to the group that received 50 mg zinc sulfate capsules thrice daily for 14 days. There was also a subjective and objective significant difference in the severity of oral mucositis, evaluated at days 4, 7 and 14 after chemotherapy. The authors recommended that zinc sulfate should be used in clinical settings as it reduced the incidence and severity of chemotherapy-induced oral mucositis [45]. Gholizadeh and colleagues, Arbabi-Kalati and colleagues as well as de Menêses also found a positive effect of zinc sulfate supplementation (ranging from 75 mg to 660 mg per day) on the reduction of oral mucositis incidence among patients suffering from acute myeloid leukemia undergoing chemotherapy in their randomized, double-blinded, placebo-controlled clinical trials. They concluded that zinc is a promising strategy for delaying the occurrence of chemotherapy-induced oral mucositis and reducing its intensity [42,46,47]. 

## 9. Role of Zinc Deficiency in Cleft Lip, Cleft Palate and Salivary Glands

Various studies have been performed to assess the eventual association of zinc deficiency and oral malformations. Researchers performed case-control studies comparing women whose babies suffered from nonsyndromic cleft lips/palates with mothers of unaffected children in the Philippines and Poland, and found that higher plasma zinc in women in reproductive age was associated with a lower risk of developing cleft lip/palate in their offspring. Copper injection and low zinc diet during pregnancy were also linked to orofacial clefts [48,49,50]. The case-series results published in 2018 supported previous results, that low plasma zinc levels of healthy mothers were associated with nonsyndromic cleft lip/palate among their babies [51]. More recently, a case-control study confirmed that exposure to a high level of zinc during pregnancy was associated with reduced risk of cleft lip and/or palate [52].

Increased zinc has been shown to affect salivary glands. A twelve-month rat study that examined the effect of administering increased levels of zinc on chronic exposure to cadmium, one of the main chemical pollutants in the developed world, found that increased zinc intake had a protective effect on the sublingual gland against oxidative damage with chronic cadmium exposure [53]. However, increased zinc intake should be monitored closely as zinc toxicity has been proven to have a negative effect on the quantity and quality of saliva, probably through changes in neurologic pathways to the salivary glands or effect on the acini cells [54]. 

## 10. Benefits of Zinc Treatment in Oral Diseases

Not only is zinc a component of many mouth rinses and toothpaste as mentioned previously, it is also incorporated in various dental materials used in dental restorations. When zinc is added to composite restorative materials, it increases its mechanical properties and significantly reduces Streptococcus mutans growth [17]. Zinc oxide eugenol, a cement mostly used as a temporary filling material or a base under a permanent filling, has been proven to be better than resin cement for using around dental implants. It does not cause peri-implant mucositis which increases periodontal pockets, especially in patients with a history of periodontitis [55,56]. 

Khajuria and colleagues performed an experimental clinical study on an animal model of periodontitis to develop a chitosan-based risedronate/zinc hydroxyapatite intrapocket dental film that could be applied in the treatment of alveolar bone loss. [57]. Their results revealed that the treatment effectively reduced alveolar bone destruction and contributed to the healing of the periodontium [57]. Photodynamic inactivation with quaternized Zn (II) phthalocyanines and specific light irradiation are effective for inactivating drug-resistant periodontal bacteria such as Enterococcus faecalis, Prevotella intermedia, and Aggregatibacter Actinomycetemcomitans [58]. A single-blinded study of adult patients with chronic periodontitis revealed that a new chemical device containing zinc and octenidine considerably reduced bacteria amounts both for some species such as Treponema Denticola and Treponema Forsythia as well as for the total bacterial count. Other benefits of the new device were improved patient compliance, better access to periodontal pockets and a lower dosage of antimicrobial agents [59]. 

Hydrogel containing zinc showed broad-spectrum antimicrobial and effective antibacterial action. In addition, it was found to be secure with low toxicity [60]. Zinc was also found to have a strong photocytotoxic antiviral activity especially on enveloped viruses such as herpes simplex type 1 and influenza virus [61,62,63]. In a case series, wherein 50 mg zinc sulfate supplements were given daily for two to three months to three cases of oral dysgeusia, symptoms were reduced after one month, leading the authors to conclude that zinc deficiency may be one of the causes of taste disorders [64].

## 11. Zinc Toxicity

Zinc toxicity can be acute or chronic. Acute zinc toxicity is a rare event as zinc is relatively harmless compared to other metals with the same properties [7]. However, patients with flu, taking zinc lozenges or liquid supplements might experience a metallic taste. The acute form is caused by exposure to high doses of zinc. The individual will present with nausea, loss of appetite, vomiting, abdominal cramps especially epigastric pain, diarrhea, headaches, lethargy and fatigue [12]. In some unusual cases, individuals working in metallurgy can breathe zinc fumes and develop a condition, called ‘metal fume fever’, which is an acute condition, and usually lasts around 24–48 h; clinically it is evident by chill, fever, sweating, weakness, coughing, muscle soreness, chest pain and shortness of breath. The chronic form of zinc toxicity occurs when the individual has been exposed to long term high intakes of zinc that reach 150 mg to 450 mg per day. Signs of chronic zinc toxicity include lethargy, copper deficiency, alteration of iron function leading to severe anemia, impaired immunity, obesity, and its related diseases, as well as reduced levels of lipoproteins [4]. Zinc toxicity increases glycosylated hemoglobin levels in the blood making diabetic patients more vulnerable to developing related complications, for instance, atrial fibrillation [9,65]. Finally, zinc toxicity is uncommon in humans [66,67]. Ingesting far more than 50 mg of zinc could induce toxicity [66]. Commercially available vitamins and nutrient supplements usually contain zinc, and simultaneously consuming multiple supplements may be of risk of higher intake of zinc than the RDA. As mentioned, in zinc toxicity, excessive absorption of zinc can diminish copper and iron absorption; such reduced absorption could induce iron deficiency, with possible clinical consequences, including peripheral neuropathy. Excessive zinc levels are cytotoxic and higher systemic zinc can cause death in experimental animals [68].

## 12. Conclusions

The aforementioned studies indicate that zinc is an essential trace element for maintaining oral health. Due to its effectiveness in oral care, it is a component of many dental materials, mouth rinses, and toothpaste. Zinc is effective in the treatment of oral mucosa diseases especially aphthous ulcer and is useful in the prevention and treatment of plaque-related diseases such as periodontal diseases and dental caries. It has also been identified as a biomarker of oral squamous cell carcinoma and should be evaluated in oral suspicious lesions. 

Zinc deficiency is widespread and can be caused by several factors. It is associated with the pathogenesis of common oral mucosa diseases as well as other diseases. Healthy nutrition rich in zinc and other micronutrients is therefore essential in optimizing good oral health. The development of specific recommendations for zinc supplementation and other micronutrients is also important to recognize the role of these micronutrients in maintaining good health. Further research studies and conducting well-designed, randomized clinical trials that focus on supplementation intending to use zinc as adjunct therapy are therefore needed.

With common knowledge of the role of zinc in human health and widespread zinc deficiency, education on the sources of zinc and the importance of consuming a diet rich in zinc is vital. Zinc supplements should be given to high-risk groups such as pregnant women, children, adolescents and the elderly, especially in developing countries. Iron supplements, which can negatively affect zinc absorption, should be taken between meals so that they do not compete with zinc for absorption. Finally, an adequate balance of vitamins and micronutrients, including zinc is important for maintaining oral health and general health [69,70,71,72,73,74,75]. 

## Figures and Tables

**Figure 1 nutrients-12-00949-f001:**
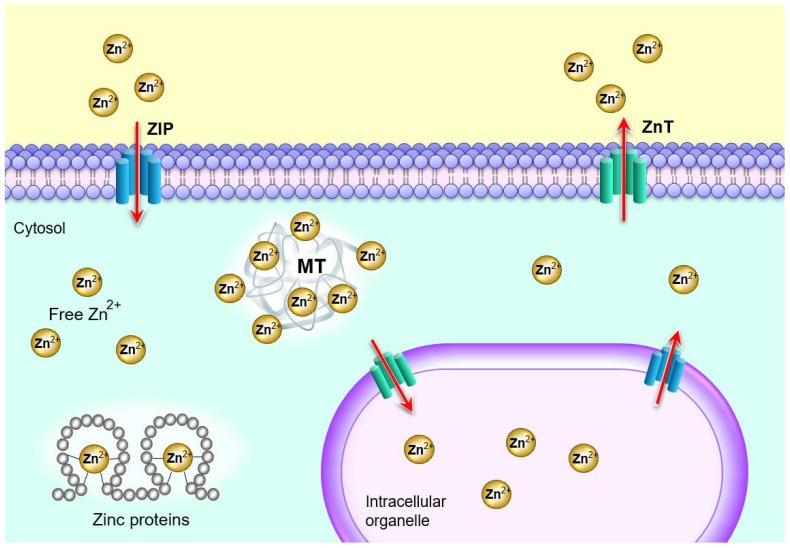
Cellular zinc homeostasis is controlled by the coordinated interactions among metallothioneins (MT), Zrt- and Irt-like proteins (ZIP), and Zn transporters (ZnT). The two zinc transporter families, ZIP and ZnT delicately control the cytosolic movement of zinc; MTs bind to zinc to reserve, buffer, and chelate [11].

**Table 1 nutrients-12-00949-t001:** List of foods containing zinc.

Categories of Foods	Types of Foods (per 100 gm)	Milligrams (mg) per Serving
**Cereals and Whole Grains**	Brown/black rice	2.2–5.9
	Black sesame	7.7
	Rye	5.0
	Oats	4.2
	Quinoa	1.0
**Meat and Poultry**	Chicken	1.6
	Lamb	4.6
	Liver	4.0
	Beef	8.4
**Seafood**	Oysters	78.6
	Crab	5.4
	Lobster	3.5
**Fruits**	Dates	0.4
	Pomegranates	0.3
	Berries	0.6
	Avocadoes	0.6
**Vegetables**	Soy foods	0.9–4.8
	Mushrooms	1.0
	Cabbage	0.2
	Spinach	0.7
	Broccoli	0.4
	Garlic	1.1
**Milk and Dairy**	Whole milk	0.3
	Yogurt	0.5
	Cheese	3.2
**Legumes**	Peas	0.8
	Lentils	4.7
	Beans	0.2–5.4
	Peanuts	2.1
	Chickpeas	3.4
	Edamame	1.3
**Nuts and Seeds**	Nuts/Cashews	5.7
	Sunflower seeds	5.3
	Almonds	3.0
	Pumpkin seeds	7.6–10.3
	Pecans	4.5
	Chia seeds	4.5

**Table 2 nutrients-12-00949-t002:** Zinc recommended daily intakes at different levels of life; Adapted from Devi et al., 2014 [4].

	Age	Zinc Recommended Daily Intakes in mg/day
**Males**	9–13 years	8
	14–18 years	11
	19–30 years	11
	31–50 years	11
	51–70 years	11
	>70 years	11
**Females**	9–13 years	8
	14–18 years	9
	19–30 years	8
	31–50 years	8
	51–70 years	8
	>70 years	8
**Females - Pregnant**	< 18 years	13
	19–30 years	11
	31–50 years	11
**Females - Lactating**	< 18 years	13
	19–30 years	11
	31–50 years	11
**Children**	1–3 years	3
	4–8 years	5
**Infants**	0–6 months	2
	7–12 months	3

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
