# Peer review of "Zinc Adequacy Is Essential for the Maintenance of Optimal Oral Health"

_nutrients, 2020, doi:10.3390/nu12040949_

Round 1
Reviewer 1 Report
The authors comprehensively summarized the published reports regarding the relationship between zinc and oral homeostasis. Aphthous ulcer is assumed as one of symptoms of zinc deficiency. However, I have never seen patients with acrodermatitis enteropathica accompanied with aphthous ulcer. In this respect, the relationship between zinc and oral homeostasis may be still controversial.
Did authors find papers in which aphthous ulcer was observed in patients with zinc deficiency such as acrodermatitis enteropathica? If not, please discuss about the controversy in the manuscript. Are there reports about the histological changes of oral epithelium and mucosa by zinc deficiency?
Author Response
The authors comprehensively summarized the published reports regarding the relationship between zinc and oral homeostasis. Aphthous ulcer is assumed as one of symptoms of zinc deficiency. However, I have never seen patients with acrodermatitis enteropathica accompanied with aphthous ulcer. In this respect, the relationship between zinc and oral homeostasis may be still controversial.
Did authors find papers in which aphthous ulcer was observed in patients with zinc deficiency such as acrodermatitis enteropathica? If not, please discuss about the controversy in the manuscript. Are there reports about the histological changes of oral epithelium and mucosa by zinc deficiency?
- We want to thank the reviewer for raising this important point, in the revised manuscript, as correctly recommended by the author, we’ve mentioned, “….it is, however, important to mention that despite the association between zinc deficiency and aphthous ulcer in human studies, whether such association with zinc inadequacy is causative or not, is an area needs further experimental validation”.
- A few publications reported the association between zinc deficiency and aphtous ulcer (references #22-25; original submission)
- A case report on two siblings with acrodermatitis enteropathica revealed that three days after their birth, there were erythematous, erosive, and crusted lesions on their mouths and tongues (Mahboobeh & Fatemeh, 2020). However, when they presented in the clinic, the oral lesions were no longuer there. They were discovered only through medical history; reports on cases of acrodermatitis enteropathica talk about glossitis and stomatitis.
Reviewer 2 Report
The present review reports the effects of Zinc on dental carries, periodontal tissues, oral malignancy, oral Mucositis, Cleft Lip, Cleft Palate and Salivary glands, as well as in oral diseases treatment. The review also lists the beneficial effects of Zinc supplementation, as its deficiency is very common.
Table 1 lists all the foods containing Zinc but it does not list which source is rich in Zinc. There may be a mix in interpretations of Table 2 and 3. In page 5, the interpretation of Table 2 does not correspond with Table 2 listing on page 4. The measured amount of Zinc should be listed as mg per body weight. The paragraph 13 on Zinc toxicity is succinct and leaves the reader wanting more reading.
Author Response
The present review reports the effects of Zinc on dental carries, periodontal tissues, oral malignancy, oral Mucositis, Cleft Lip, Cleft Palate and Salivary glands, as well as in oral diseases treatment. The review also lists the beneficial effects of Zinc supplementation, as its deficiency is very common.
Table 1 lists all the foods containing Zinc but it does not list which source is rich in Zinc. There may be a mix in interpretations of Table 2 and 3. In page 5, the interpretation of Table 2 does not correspond with Table 2 listing on page 4. The measured amount of Zinc should be listed as mg per body weight. The paragraph 13 on Zinc toxicity is succinct and leaves the reader wanting more reading.
- We want to thank the reviewer for kindly pointing out the error in Table leveling, we fix the error, & as recommended by Reviewer 3, we’ve deleted Table 3. As suggested, we’ve listed the measured amount of Zinc as mg per body weight. As correctly requested, we’ve elaborated on the zinc toxicity section.
Reviewer 3 Report
This review paper describes the role of zinc and zinc supplementation in oral health. It is an important topic with high relevance to public health. There are several suggestions below that would make this review useful to clinicians and other public health professionals.
- Throughout the manuscript it is important to distinguish between the roles of normal amounts of zinc (from food or supplements) and the role of zinc supplementation above and beyond the usual requirements. For example, Section 5 has the statement “In the oral cavity, zinc is naturally present in various sites such as saliva, dental plaque, and dental hard tissues.” This very reasonable suggests that normal levels of dietary zinc are needed to maintain these normal functions. However, the next sentence says, “Consequently, it is effective against different oral diseases including dental caries, gingivitis, periodontitis, oral cancer, cleft lips/palate and foul odor [17-19].” It is not clear what is meant by “it is effective”. Does this mean that supplementation is effective? If so, what is the evidence for each of these? How much supplementation is needed to provide the hypothesized protective effect? Or does this simply mean that normal amounts of zinc are needed to prevent the development of these disorders? Again, the evidence (supporting data) for this needs to be presented.
- Another example of the above is the statement in section 9 “In addition, zinc has been found to prevent the transformation of the lesion into malignancy”. Does this mean supplementation of zinc does this? Or does it just mean that we need adequate zinc? In any case, what is the actual evidence to support this statement?
- Please include the amount of zinc that was used and the length of the treatment for any references to zinc supplementation (particularly in humans) throughout this manuscript. For example, in section 6 there is the statement, “In a double-blinded placebo-controlled study zinc sulfate treatment was found to have a much more rapid and sustainable action on aphthous ulcer than dapsone [26].” There are many other examples that need to be updated.
- More analysis of the data is needed. For example, Section 9 states that it was “found that zinc level was lower in patients with oral submucous fibrosis. The authors concluded that alongside clinical features, zinc may be an additional biomarker for development, diagnosis and prognosis of oral submucous fibrosis and other premalignant lesions.” Later in this section there is the statement that “The level of zinc was found to be low in patients with oral squamous cell carcinoma”. An alternative interpretation for both of these is that zinc deficiency is simply an effect of reduced food intake in patients with oral cancers.
- The introduction states that “the optimum level of zinc is <100mg/day”. This is incorrect. Zinc is classified as a trace mineral because it is falls into the category of minerals that are needed in quantities less than 100mg/day. But this does not mean that this is the optimum amount. In fact, the upper limit for zinc is 40mg/day.
- Table 1 would be more useful if it gave amounts of zinc in a typical serving of these foods.
- Table 3 is not needed because all of this information is listed in the text.
Minor Editorial Issues
- Inconsistent use of capitalization of zinc (Zinc vs zinc). The word zinc should not be capitalized unless it is at the beginning of a sentence.
- The introduction states that “Clinical trials have revealed that zinc can contribute to a decrease in the demineralization of enamel [2].” This sounds like zinc causes demineralization. Do you mean zinc deficiency leads to demineralization?
- The first two sentences of section 5 are redundant (“Many studies performed on zinc as a nutritional factor found that it has great effect on oral tissues especially oral mucosa and periodontium [16]. Zinc has been proven to be an essential trace element for oral health maintenance.”).
Author Response
This review paper describes the role of zinc and zinc supplementation in oral health. It is an important topic with high relevance to public health. There are several suggestions below that would make this review useful to clinicians and other public health professionals.Throughout the manuscript it is important to distinguish between the roles of normal amounts of zinc (from food or supplements) and the role of zinc supplementation above and beyond the usual requirements. For example, Section 5 has the statement “In the oral cavity, zinc is naturally present in various sites such as saliva, dental plaque, and dental hard tissues.” This very reasonable suggests that normal levels of dietary zinc are needed to maintain these normal functions. However, the next sentence says, “Consequently, it is effective against different oral diseases including dental caries, gingivitis, periodontitis, oral cancer, cleft lips/palate and foul odor [17-19].” It is not clear what is meant by “it is effective”. Does this mean that supplementation is effective? If so, what is the evidence for each of these? How much supplementation is needed to provide the hypothesized protective effect? Or does this simply mean that normal amounts of zinc are needed to prevent the development of these disorders? Again, the evidence (supporting data) for this needs to be presented.
- We want to thank the reviewer for carefully reading our manuscript & providing useful suggestions. We have distinguished between the roles of normal zinc amounts and the role of zinc supplementation throughout the manuscript & provided evidence to make it clearer for the reader.
- Another example of the above is the statement in section 9 “In addition, zinc has been found to prevent the transformation of the lesion into malignancy”. Does this mean supplementation of zinc does this? Or does it just mean that we need adequate zinc? In any case, what is the actual evidence to support this statement?\
- The statement in section 9 has been re-written in more detail to provide clarity for the reader.
- Please include the amount of zinc that was used and the length of the treatment for any references to zinc supplementation (particularly in humans) throughout this manuscript. For example, in section 6 there is the statement, “In a double-blinded placebo-controlled study zinc sulfate treatment was found to have a much more rapid and sustainable action on aphthous ulcer than dapsone [26].” There are many other examples that need to be updated.
- The amount of zinc used and the length of treatment for references to zinc supplementation, especially in humans, have now been added throughout the manuscript.
- More analysis of the data is needed. For example, Section 9 states that it was “found that zinc level was lower in patients with oral submucous fibrosis. The authors concluded that alongside clinical features, zinc may be an additional biomarker for development, diagnosis and prognosis of oral submucous fibrosis and other premalignant lesions.” Later in this section there is the statement that “The level of zinc was found to be low in patients with oral squamous cell carcinoma”. An alternative interpretation for both of these is that zinc deficiency is simply an effect of reduced food intake in patients with oral cancers.
- This section has been expanded to provide more analysis of the data observed by the studies & explanations have been provided for data obtained.
- The introduction states that “the optimum level of zinc is <100mg/day”. This is incorrect. Zinc is classified as a trace mineral because it is falls into the category of minerals that are needed in quantities less than 100mg/day. But this does not mean that this is the optimum amount. In fact, the upper limit for zinc is 40mg/day.
- The sentence has been replaced to give the RDA of zinc depending on different stages of life & sex. We have also included a statement that the upper limit for zinc is 40 mg/day.
- Table 1 would be more useful if it gave amounts of zinc in a typical serving of these foods.
- The amount of zinc in servings of 100 g for the various food items has been provided. We want to thank the reviewer for this important suggestion.
- Table 3 is not needed because all of this information is listed in the text.
- We’ve deleted Table 3 from the revised manuscript.
Minor Editorial Issues
- Inconsistent use of capitalization of zinc (Zinc vs zinc). The word zinc should not be capitalized unless it is at the beginning of a sentence.
- We want to thank the reviewer for kindly pointing out the error, we’ve fixed the error throughout the manuscript.
- The introduction states that “Clinical trials have revealed that zinc can contribute to a decrease in the demineralization of enamel [2].” This sounds like zinc causes demineralization. Do you mean zinc deficiency leads to demineralization?
- Again, we want to thank the reviewer for carefully reading our manuscript, we’ve rewritten the part to convey that zinc deficiency leads to demineralization in the revised manuscript. The sentence has been changed to “Clinical trials have demonstrated that zinc ions decrease the rate of enamel demineralization [2].”
- The first two sentences of section 5 are redundant (“Many studies performed on zinc as a nutritional factor found that it has great effect on oral tissues especially oral mucosa and periodontium [16]. Zinc has been proven to be an essential trace element for oral health maintenance.”).
- As correctly recommended, we’ve erased the redundant section from the revised manuscript.